# Inference on a Multi-Patch Epidemic Model with Partial Mobility, Residency, and Demography: Case of the 2020 COVID-19 Outbreak in Hermosillo, Mexico

**DOI:** 10.3390/e25070968

**Published:** 2023-06-22

**Authors:** Albert Orwa Akuno, L. Leticia Ramírez-Ramírez, Jesús F. Espinoza

**Affiliations:** 1Departamento de Probabilidad y Estadística, Centro de Investigación en Matemáticas A.C., Jalisco s/n, Colonia Valenciana, Guanajuato C.P. 36023, Gto, Mexico; leticia.ramirez@cimat.mx; 2Departamento de Matemáticas, Universidad de Sonora, Rosales y Boulevard Luis Encinas, Hermosillo C.P. 83000, Sonora, Mexico; jesusfrancisco.espinoza@unison.mx

**Keywords:** epidemics, human mobility, inference, deterministic inversion, Bayesian inference

## Abstract

Most studies modeling population mobility and the spread of infectious diseases, particularly those using meta-population multi-patch models, tend to focus on the theoretical properties and numerical simulation of such models. As such, there is relatively scant literature focused on numerical fit, inference, and uncertainty quantification of epidemic models with population mobility. In this research, we use three estimation techniques to solve an inverse problem and quantify its uncertainty for a human-mobility-based multi-patch epidemic model using mobile phone sensing data and confirmed COVID-19-positive cases in Hermosillo, Mexico. First, we utilize a Brownian bridge model using mobile phone GPS data to estimate the residence and mobility parameters of the epidemic model. In the second step, we estimate the optimal model epidemiological parameters by deterministically inverting the model using a Darwinian-inspired evolutionary algorithm (EA)—that is, a genetic algorithm (GA). The third part of the analysis involves performing inference and uncertainty quantification in the epidemic model using two Bayesian Monte Carlo sampling methods: t-walk and Hamiltonian Monte Carlo (HMC). The results demonstrate that the estimated model parameters and incidence adequately fit the observed daily COVID-19 incidence in Hermosillo. Moreover, the estimated parameters from the HMC method yield large credible intervals, improving their coverage for the observed and predicted daily incidences. Furthermore, we observe that the use of a multi-patch model with mobility yields improved predictions when compared to a single-patch model.

## 1. Introduction

Research on the mathematical modeling of infectious diseases has witnessed continuous growth in recent decades, primarily motivated by the substantial impacts that contagious diseases have had in various crucial domains, including social, health, and economic development [1,2,3,4,5]. These domains are intricately linked to interactions within the human population, which serve as a key driving factor of disease.

Indeed, works such as [6,7] have raised awareness of the fact that the emergence and re-emergence of infectious diseases have been historically neglected as a significant threat to public health. Therefore, understanding the propagation dynamics of infectious diseases has become a central issue for epidemiologists and theoreticians [8,9,10,11,12]. Such knowledge subsequently enables the formulation of effective infection mitigation policies which, in turn, contribute to the maintenance of a healthy human population, thereby significantly bolstering the economic development of their respective nations.

Since its advent and declaration as a pandemic in March 2020 by the WHO, COVID-19 has undeniably wreaked havoc and drastically impacted the lives and livelihoods of the global human population. In terms of significance, COVID-19 has been compared to the influenza pandemic of 1918–1920 [8]; however, these pandemics have not yet been equated in terms of severity. The 1918 influenza pandemic had an estimated death toll of 40 million lives and infected one-third of the world’s population. In comparison, to date, COVID-19 has had a death toll of 6.6 million and 633 million confirmed cases (accounting for less than 10% of the total human population) [13]. However, the urgent need to conduct further research on COVID-19 and other infectious diseases is underscored by the warnings issued by scientists in recent years regarding the imminent threat of another pandemic. These cautionary warnings emphasize the pressing need to proactively study and understand these diseases to be better prepared for any potential future outbreaks [14]. These studies seek to understand the dynamics of infection propagation, control, and mitigation strategies. They encompass various aspects, including (but not limited to) model selection, statistical inference, uncertainty quantification, and prediction using empirical epidemiological data. Such studies provide epidemiologists and policymakers with valuable information regarding the effectiveness of control and eradication measures. This knowledge facilitates the formulation of proactive response strategies in anticipation of a potential new pandemic crisis.

Human behavior and interactions are at the core of the transmission dynamics of infectious and communicable diseases such as COVID-19. Consequently, researchers and epidemiologists involved in the study of infectious diseases must accurately incorporate human behavior as a critical component of the models used for modeling infectious diseases [15]. One significant human behavior that has consistently influenced the person-to-person transmission of infectious diseases during outbreaks is mobility [10,16,17,18,19,20,21]. Human mobility and travel patterns promote the creation of temporal social connections and networks that—whether local or global—play a crucial role in the spread of diseases. Human mobility, an undeniably crucial component of human existence and survival, not only plays a critical role in the spread and propagation of epidemics [18,22,23,24], but also crucially determines the speed of the spread of infectious diseases [23,25,26]. Therefore, it cannot be denied that a deep understanding of the role of human mobility in the spread of infectious diseases is central in the quest to have a clear view, for instance, of the impact of control strategies such as mobility restrictions [27,28,29,30,31].

Developments in epidemiological research have confirmed that traditional compartmental epidemic models are deficient in modeling strong heterogeneous disease spread, as they are formulated with the assumption of interactions within a well-mixed homogeneous population [15,32,33]. It has been widely recognized that humans do not exhibit homogeneous mobility patterns [15], highlighting the importance of incorporating the behavioral responses of a heterogeneous population into epidemiological models, as emphasized by [34]. The scientific literature offers a wide array of proposed models that aim to capture the heterogeneous mobility and dispersal patterns observed among humans. Many of these models consist of human groups of spatially separated populations that interact at some level (representing a geo-meta-population multi-patch framework), as evidenced by numerous studies in the literature (see, e.g., [26,27,34,35,36,37,38,39,40,41,42]). Although modeling the geospatiotemporal spread of infectious diseases is a complex task [43], models such as those in the mentioned references have crucially attempted to accurately capture human mobility in different settings. Nevertheless, there is still a need to incorporate further realistic mobility patterns that reflect more complex mobility scenarios [17], which is an inevitable reality when considering human behavior. In certain scenarios, it may be highly desirable to explore the relationship between the spread of infectious diseases and the proportion of time individuals spend in different locations during their travels [44]. This approach considers the impact of human movement on disease transmission and how it relates to the amount of time spent in different locations, with each having specific characteristics, such as the level of risk of infection. These locations, which are homogeneous within themselves and potentially differ between themselves, are usually referred to as patches in the literature. Then, a single-patch model can be reduced to a model with a well-mixed homogeneous population, where every susceptible individual can become infected with the same probability. The concept of multiple patches allows locations with different characteristics to be considered, helping to capture the spatial dynamics and interactions produced by human mobility that influence the spread of infectious diseases.

Once an appropriate epidemic model has been formulated, a myriad of analyses relevant to epidemiologists, policymakers, and public health officials can be conducted. These include (but are not limited to) theoretical analysis and numerical simulations (see, e.g., [39,40,45,46,47]), inference, uncertainty quantification, and predictions using empirical data (see, e.g., [14,48,49,50,51,52,53]). Most studies modeling population mobility and the spread of infectious diseases, particularly those using meta-population multi-patch models, have tended to focus on the theoretical and global properties and numerical simulations of such models. Relatively few studies have focused on inference and uncertainty quantification in epidemic models including population mobility. To the best of our knowledge, to date, the only study that has come close to performing inference, uncertainty quantification, and predictions on a multi-patch epidemic model is [51], in which a projected Stein Variational Gradient Descent (pSVGD) algorithm was used to model COVID-19 in the states of New Jersey and Texas in the U.S.; however, their model did not incorporate population mobility, as they modeled the dynamics and severity of COVID-19 inside and outside long-term care (LTC) facilities, where the interaction was introduced through a contact matrix. Other works that have addressed the problem of statistical inference for epidemics on networks include [54,55]. These models consider families of networks of contact that remain constant over time. Filling the gap left by the scarce literature on uncertainty quantification and Bayesian inverse problems on meta-population multi-patch epidemic models is the central motivation of the present study. To achieve this objective, we conduct inference and uncertainty quantification using confirmed COVID-19 cases in Hermosillo, Mexico, and employ a multi-patch epidemic model incorporating mobility, residency, and demography aspects. This model was first proposed and theoretically analyzed in [42].

Solving an inverse problem entails the utilization of both observed data and a model to deduce the parameter values that characterize the dynamics producing the observations [56,57,58]. However, employing models that adequately capture the heterogeneity and complexity of human behavior, such as mobility, in modeling the spread of infectious diseases introduces numerous parameters [51,59]. This, in turn, presents a multitude of challenges when addressing inverse problems (fit) and conducting inference (uncertainty quantification). To confront the problems originating from the complex system and its high-dimensional parameter space, we divide the problem into two parts: the first is related to the mobility parameters that describe the proportion of individuals who move and the time spent in each of the sub-regions/patches, while the second is associated with the numerical fit or statistical inference of the initial conditions and parameters for the infectious agent, as well as the individual evolution of infected individuals.

The remainder of this article is organized as follows: In Section 2, we present and describe the epidemic model on which the present analysis is centered. Section 3 discusses how the COVID-19 data used in this work were obtained. In Section 4, a discussion on the estimation of the mobility parameters and residence times of the epidemic model using mobile phone sensing data and the Brownian bridge model is presented. We elucidate the formulation of the deterministic inversion model and the corresponding results in Section 5. Section 6 details the Bayesian model formulation, inference, and the obtained results. Finally, we provide concluding remarks and possible extensions of the present work in Section 7.

## 2. The Multi-Patch Model with Mobility, Residency, and Demography

The emergence and re-emergence of infectious diseases over the past several decades has led to the formulation of numerous mathematical models incorporating varied and distinct disease propagation dynamics by researchers and epidemiologists. The central purpose of such models is to provide fundamental quantitative information and utilitarian guidelines for disease outbreak management and mitigation policy formulations. The provision of such useful information by these models is based on their ability to incorporate and capture realistic situations and disease transmission parameters. One such parameter which plays a crucial role in the spread of infectious diseases is human population mobility [16,17,18,21,39,40,60].

Several aspects of human history, such as social and economic factors, have been significantly impacted by the effects of the spread of infectious disease viruses among mobile sectors of human civilization [21]. Undoubtedly, the effects of the current COVID-19 pandemic is a demonstration of the devastating consequences of the interchange between human population mobility and infectious disease transmission. For instance, in December of 2019, Wuhan, China was the only understood geographical boundary of COVID-19 [61,62,63,64,65]. However, it can be postulated that human mobility within and between cities in China and around the world aided the spread of the pathogen to a significant portion of the planet. In a span of three months, the pathogen had posed monumental health, social, and economic stresses, to the point of being declared a pandemic by the WHO on 11 March 2020. Even though the exact nature of the contagion remained unclear, the undeniable interplay between the swift spread of the virus and human mobility through universally known modes of travel was recognized [16,66,67,68,69]. Indeed, besides aiding the spread of COVID-19, the close relationship between population mobility and the spread of infectious diseases has the potential to shape the emergence of future epidemics. Consequently, in order to formulate robust guidelines to manage both the current ongoing COVID-19 pandemic and possible future pandemics, there is dire need to construct appropriate epidemic models that accurately capture population mobility behaviors.

Human mobility, being a geospatiotemporal phenomenon [17,43,70,71], can be better modeled using meta-population multi-patch models [4,34,39,40,42,44], as traditional homogeneous compartmental models are incapable of capturing such a strong heterogeneous human behavior [15,17,34,51]. Once such models have been constructed, a wide spectrum of quantitative analyses leading to a deeper understanding of the relationship between mobility and the spread of infectious diseases can be conducted. For instance, researchers and epidemiologists may focus on theoretical properties and numerical simulations of the model or parameter estimation, uncertainty quantification, and predictions. In this research, we focus on the latter, using confirmed COVID-19 cases in Hermosillo, Mexico and a multi-patch SEIRS compartmental epidemic model with residency and demography characteristics. The model was proposed and its theoretical properties were analyzed in [42]. It is crucial to emphasize that, in this model, human mobility encompasses more than just travel between two locations; in particular, it involves the multiple temporal interactions that individuals experience while traveling to different destinations.

The following ordinary differential Equations (Equation 1) constitute the full form of the considered multi-patch non-linear model. This model extends the traditional compartmental SEIR model, which divides the population into different compartments based on their disease status: Susceptible (S), Exposed (E), Infected (I), and Recovered (R). Most individuals start in the susceptible compartment (S) but, when one of them comes into contact with an infectious individual, they move from the susceptible compartment into the exposed compartment (E), meaning that they have been infected but are not yet infectious. This individual leaves the E compartment to enter the I compartment once they can transmit the disease to susceptible individuals. Over time, infected individuals either recover from the disease or succumb to it. Once individuals recover, they move to the recovered compartment (R) and gain immunity to the disease. In the proposed model, the population dynamic also considers births (into the S compartment) and deaths (not produced by the disease).

The parameters of model (Equation 1) are described in Table 1. We refer the reader to [42] for a full description of the arguments leading to the formulation of this model.
(1)$$\left\{\begin{array}{cc} {\dot{S}}_{i} & =\;\Lambda_{i}-\beta_{i}\left(1-\alpha_{i}\right)S_{i}\frac{\left(1-\alpha_{i}\right)I_{i}+\sum_{k=1}^{n}{\tilde{p}}_{ki}I_{k}}{\left(1-\alpha_{i}\right)N_{i}+\sum_{k=1}^{n}{\tilde{p}}_{ki}N_{k}}-\sum_{j=1}^{n}\left(\beta_{j}{\tilde{p}}_{ij}S_{i}\frac{\left(1-\alpha_{j}\right)I_{j}+\sum_{k=1}^{n}{\tilde{p}}_{kj}I_{k}}{\left(1-\alpha_{j}\right)N_{j}+\sum_{k=1}^{n}{\tilde{p}}_{kj}N_{k}}\right)\\ & -\mu_{i}S_{i}+\tau_{i}R_{i}\\ {\dot{E}}_{i} & =\;\beta_{i}\left(1-\alpha_{i}\right)S_{i}\frac{\left(1-\alpha_{i}\right)I_{i}+\sum_{k=1}^{n}{\tilde{p}}_{ki}I_{k}}{\left(1-\alpha_{i}\right)N_{i}+\sum_{k=1}^{n}{\tilde{p}}_{ki}N_{k}}+\sum_{j=1}^{n}\left(\beta_{j}{\tilde{p}}_{ij}S_{i}\frac{\left(1-\alpha_{j}\right)I_{j}+\sum_{k=1}^{n}{\tilde{p}}_{kj}I_{k}}{\left(1-\alpha_{j}\right)N_{j}+\sum_{k=1}^{n}{\tilde{p}}_{kj}N_{k}}\right)\\ & -\left(\kappa_{i}+\mu_{i}\right)E_{i}\\ {\dot{I}}_{i} & =\kappa_{i}E_{i}-\left(\gamma_{i}+\psi_{i}+\mu_{i}\right)I_{i}\\ {\dot{R}}_{i} & =\;\gamma_{i}I_{i}-\left(\tau_{i}+\mu_{i}\right)R_{i}, \end{array}\right.$$
where ${\tilde{p}}_{kj}=\alpha_{k}p_{kj}$, i=1,2,⋯,n, and *n* is the number of patches.

Model (Equation 1) is limited to infectious diseases with single-strain mutation; that is, in the formulation of the model [42], it is assumed that the recovered individuals in each patch are conferred with partial immunity, which they lose at a rate of $\tau_{i}$. These individuals then re-enter the susceptible compartment, but it only in relation to the same strain of the pathogen. Notably, many infectious diseases—including COVID-19, which is the focus of this research—have several pathogenic strains whose dynamic properties should be studied for complete sensible management of emerging and re-emerging outbreaks with multiple variants. To this end, model (Equation 1) can be improved to include multiple COVID-19 disease strains.

Works such as [72,73,74,75,76] have considered models that incorporate several disease strains and mutations, with some of them focusing on this property with respect to the COVID-19 disease. Even though these works are mostly single-patch based, they provide valuable insights and serve as a useful starting point to extend model (Equation 1) in this important direction.

## 3. Data

Based on an institutional agreement between the University of Sonora (UNISON) and the Secretaría de Salud del Estado de Sonora (SSA), a collaborative effort was undertaken to provide accurate and reliable information to the population of Sonora during the pandemic emergency. As part of this initiative, researchers from the Mathematics Department at UNISON were granted access to georeferenced data pertaining to COVID-19 cases in Hermosillo, Mexico. The data set encompasses the period from 1 January 2020 to 6 September 2020. It is important to emphasize that the usage of these data are strictly limited to academic activities in accordance with UNISON’s agreements with public institutions, and utmost care has been taken to safeguard sensitive and private information.

Furthermore, on 14 September 2020, UNISON entered into a specific agreement (referenced as 12615-5700000-000412) to provide statistical consulting services to LUMEX CONSULTORES, S.C. Esteemed research professors from the Mathematics Department (including the co-author J.F.E.) provided these consulting services. The data set provided by LUMEX CONSULTORES, S.C. comprises a comprehensive collection of approximately 80 million records, each containing the GPS position and time stamp of nearly 300,000 devices. These devices were geographically located within Hermosillo city between 21 September 2020 and 15 November 2020. The usage of these data were explicitly authorized for non-profit and academic activities, with appropriate credits duly acknowledged. Notably, user privacy was carefully protected through the meticulous anonymization of the data, employing distinct alphanumeric IDs for each individual device.

In a separate study, Ramirez et al. (2022) utilized mobile phone GPS data from 21 September 2020 to 15 November 2020 to estimate mobility parameters and residence times in the 582 urban AGEBs (Basic Census Geographical Units) of Hermosillo [4]. In alignment with our research objectives of modeling the number of COVID-19 cases in four key zones within Hermosillo, considered as aggregate AGEBs as illustrated in Figure 1, we employed the geo-referenced and GPS information to obtain the number of COVID-19 cases and estimate mobility and residence patterns within and between these zones.

The two geo-referenced databases utilized in this study did not exhibit a temporal overlap, thereby limiting the feasibility of employing the complex model (Equation 1), which relies on the estimates of numerous epidemiological parameters. To address this problem, we elected to use the weighted global COVID-19 data in terms of zonal proportions.

The global confirmed positive COVID-19 cases in Hermosillo (Figure 2) can be obtained from [77], a website managed by the Consejo Nacional de Ciencias y Technología (CONACYT) since 26 February 2020. We used the weighted global cases by considering the zonal proportions of the global COVID-19 data, a selection justified in Figure 3, from which we can observe that the proportions of the available zonal COVID-19 data were approximately stable from June 2020. Therefore, we can consider that the data present the same behavior within the period for which the mobility parameters and mobility residence times were estimated. The global (Figure 2) and derived zonal COVID-19 cases, however, exhibited high variability. Therefore, we smoothed the data using 7-day moving averages, using the smoothed version for the rest of the analysis.

Most models that have been proposed to analyze and predict the incidence of COVID-19 since the emergence of SARS-CoV-2 assume that all infected individuals are observed, which is an inexact assumption due to under-reporting of disease incidence statistics. Therefore, this assumption undermines the accuracy of the models and their predictions. Emerging diseases which present a large fraction of asymptomatic pathogen carriers, such as COVID-19, Typhoid fever, Hepatitis B, Epstein–Barr virus, and Zika, are often characterized by incidence under-reporting during disease surveillance [78,79,80,81]. Coupled with asymptomatic and sub-clinical carriers—a phase that is prevalent with COVID-19 disease [82,83,84,85]—another source of under-reporting of disease incidence is a lack of systematic testing. Therefore, it is necessary to account for under-reporting when modeling and fitting COVID-19 disease incidence, as failure to do so will lead to underestimation of the epidemiological characteristics of the disease, especially the transmission parameter [86].

In modeling the observed COVID-19 incidence for eight American countries, ref. [86] noted an under-reporting of COVID-19 cases in Mexico by a factor of 15. This acute case identification problem corroborated the observation in [87], where Mexico had one of the lowest numbers of COVID-19 tests performed per reported cases. We believe that this under-reporting has cascaded down to the local states and cities of Mexico, including Hermosillo. In order to account for this under-reporting, we inflated the smoothed observed COVID-19 incidence (obtained from the weighting procedure previously explained) by a factor of 15 to obtain the daily incidence data. We subsequently considered the inflated data (see Figure 4) as the true observed daily COVID-19 incidence in Hermosillo, Mexico, and used them for inference in this research.

In addition to using real count data for model fitting and uncertainty quantification, we also utilized simulated incidence data generated from the model. Our goal was to assess the identifiability of the parameters and initial conditions for the inverse problem, as well as the statistical inference performance of the model. To this end, we used epidemiological parameters of COVID-19 gathered from the literature, as presented in Table 2, as well as those simulated using the considered model (Equation 1), to obtain daily observed incidences for each zone in Hermosillo. For this simulation, we used epidemiological parameter values close to those used by [88], who modeled the COVID-19 lockdown relaxation period in Hermosillo. The parameters and initial condition values used in the simulation can be seen in Table 3; the latter were based on the total population of each zone. After the simulation of the zonal incidence using the model, we added a normal noise $e_{t}∼N\left(I_{i}\left(t\right),\sigma =100\right)$ to introduce variability into the model data. The final simulated incidence data for each zone in Hermosillo are presented in Figure 5.

## 4. Estimation of the Mobility and Residence Time Parameters

For the present study, we sought to solve an inverse problem and perform uncertainty quantification based on system (Equation 1), using confirmed COVID-19 cases for 2020 in Hermosillo, Mexico. Such data may not be sufficient for estimating all of the mobility and infection parameters, as well the initial conditions of the system. Moreover, simultaneous estimation of several parameters often leads to parameter non-identifiability problems [109,110,111,112,113,114,115]. As such, we used the mobile phone sensing data detailed in [4] for the purpose of estimating the mobility vector $\alpha_{i}$, i=1,2,⋯,n and the mobility residence time matrices $P=(p_{ij})$, i=1,2,⋯,n, j=1,2,⋯,n, which we incorporated into system (Equation 1). Using mobile consecutive *pings*, ref. [4] estimated the mobility parameter vectors and mobility residence time matrices for 582 urban AGEBs in Hermosillo, Mexico for three periods, each divided into two parts. We used the same GPS *pings* during the *Third Period, First Part* (i.e., 21 September 2020 to 11 October 2020). Using this information, we estimated the mobility and residence time matrices for the four zones in Hermosillo. As we are interested in evaluating the forecasting (after November 2020) ability of the model, we selected this period as its start date was closer to the last period of the considered COVID-19 cases. We note that we could also use mobility data from other periods and areas, employing a similar procedure to that described below, in order to estimate mobility and residence times for the four zones with respect to the period and area of interest.

Solving the inverse problem and performing uncertainty quantification on system (Equation 1) for a set total number of AGEBs (patches) posed a logical set-up challenge. Thus, using well-defined geographical demarcations of Hermosillo, we grouped the 582 AGEBs into four main zones. These zones, together with their corresponding total population, are shown in Figure 1. The mobility parameter vector $\alpha_{i}$ and residence times $p_{ij}$, i=1,2,3,4, j=1,2,3,4 were estimated by searching for individuals who activated at least 11 pings and left their AGEBs within and between the zones. For the individual AGEBs, ref. [4] estimated the probability density functions at position *z* for any individual originating from AGEB/patch *r* as
(2)$${\hat{h}}_{r}\left(z\right)=\frac{1}{n_{r}}\sum_{j=1}^{n_{r}}h_{rj}\left(z;{\hat{\sigma }}_{rj},\delta \right),$$
where $n_{r}$ is the patch population size, $h_{rj}$ is the density of the expected time of the *j*-th individual originating from the *r*-th patch, ${\hat{\sigma }}_{rj}$ is the maximum likelihood estimate of the individual standard deviation of the mobility of the *j*-th resident from the *r*-th patch, and $\delta^{2}$ is the variance of the geolocation error. After grouping and renaming the composing AGEBs of each of the four zones, we have
(3)$$n_{{}_{1}}^{\left[l\right]}{\hat{h}}_{{}_{1}}^{\left[l\right]}\left(z\right)=\sum_{j=1}^{n_{{}_{1}}}h_{1j}^{\left[l\right]}\left(z;{\hat{\sigma }}_{1j}^{\left[l\right]},\delta \right),\cdots ,n_{{}_{n_{l}}}^{\left[l\right]}{\hat{h}}_{{}_{n_{{}_{l}}}}^{\left[l\right]}\left(z\right)=\sum_{j=1}^{n_{{}_{1}}}h_{n_{{}_{l}}j}^{\left[l\right]}\left(z;{\hat{\sigma }}_{n_{{}_{l}}j}^{\left[l\right]},\delta \right),$$
where the subscript index [l] indicates that the said quantity is in zone *ℓ*, and $n_{l}$ represents the total number of AGEBs in the $l^{\mathrm{th}}$ zone.

From (Equation 3), we can obtain the estimated probability density function at location *z* for any individual originating from patch *r* in zone *ℓ* as
(4)$${\hat{h}}_{{}_{l}}^{*}\left(z\right)=\frac{1}{N^{\left[l\right]}}\sum_{r=1}^{n_{{}_{l}}}\sum_{j=1}^{n_{r}}h_{rj}^{\left[l\right]}\left(z;{\hat{\sigma }}_{rj}^{\left[l\right]},\delta \right),$$
where $N^{\left[l\right]}=\sum_{j=1}^{n_{r}}n_{j}^{\left[l\right]}$. Then, the expected occupation time in *A* of a resident of zone *ℓ* can thus be obtained, from (Equation 4), as
(5)$${\hat{P}}_{{}_{l}}\left(A\right)=\int_{A}{\hat{h}}_{{}_{l}}^{*}\left(z\right)dz=\frac{1}{N^{\left[l\right]}}\int_{A}\sum_{r=1}^{n_{{}_{l}}}\sum_{j=1}^{n_{r}}h_{rj}^{\left[l\right]}\left(z;{\hat{\sigma }}_{rj}^{\left[l\right]},\delta \right)dz.$$

As mentioned above, for this research, we considered the mobility parameter and the mobility residence matrix for the *Third Period, First Part* (21 September 2020 to 11 October 2020), which we computed (using the above procedure) to obtain
$$\alpha ={\left(\begin{array}{cccc} 0.9668, & 0.9265, & 0.9692, & 0.9680 \end{array}\right)}^{⊤}\;\mathrm{and}\;P=\left(\begin{array}{cccc} 0.8164 & 0.1289 & 0.0372 & 0.0175\\ 0.1222 & 0.8119 & 0.0215 & 0.0444\\ 0.0722 & 0.0504 & 0.7293 & 0.1481\\ 0.0313 & 0.1166 & 0.1278 & 0.7243 \end{array}\right).$$

## 5. Deterministic Inversion

Due to their exponential growth in time, the state variables of the dynamical system (Equation 1) are extremely sensitive to the input parameters. This sensitivity introduces uncertainty into the pre-inferred parameter values which, in turn, may result in divergence or significant uncertainty in the model outputs. For a robust and desirable model analysis, it is imperative to avoid such model divergences by first determining the optimal model parameters that fit the measured data. Moreover, prior knowledge of point parameter estimates is necessary for meaningful uncertainty quantification. To this end, we began by deterministically inverting the system (Equation 1), as described in this section, and then subsequently performed uncertainty quantification on the inferred point parameters, described in the following section. An outline of the formulation of the inversion problem and the results is provided below.

### 5.1. Formulation of the Model to Minimize

Let $x=\left(S,E,I,R\right)\in {\left(L^{2}\left(0,T\right)\right)}^{4n}$ where T>0, $S={\left(S_{1},\cdots ,S_{n}\right)}^{⊤}$, $E={\left(E_{1},\cdots ,E_{n}\right)}^{⊤}$, $I={\left(I_{1},\cdots ,I_{n}\right)}^{⊤}$, and $R={\left(R_{1},\cdots ,R_{n}\right)}^{⊤}$ denote the state variables in system (Equation 1) for *n* patches. Let $m_{i}$ be the dimension of the parameters to be estimated for patch *i*. Then, we have the total number of parameters to be estimated for patch *i*, denoted as $\theta_{i}=\left\{\theta_{i1},\cdots ,\theta_{im_{i}}\right\}$. Consequently, the total number of parameters, $m=\sum_{i=1}^{n}m_{i}$, for the global system is $\theta_{}={⋃}_{i=1}^{n}\theta_{i}\in R_{+}^{m}$. System (Equation 1) can be viewed as a mapping Ψ(θ)=x, where $\Psi :R_{+}^{m}⟶{\left(L^{2}\left(0,T\right)\right)}^{4n}$ (i.e., Ψ maps the parameters to be estimated to the state variables) defining an initial value problem of the form
(6)$$\dot{x}=f\left(x,\theta \right),\;x\left(t_{0}\right)=x_{0}\in R^{n},\;t_{0}\in R.$$

The function *f* in the forward problem (Equation 6) is continuous and satisfies the Lipschitz condition with respect to x (see [42]). Thus, given θ, the forward problem (Equation 6) has a unique solution x.

Besides the numerically simulated states x, the inverse problem requires directly measured state variables at a discrete set of points $t_{1},t_{2},\cdots ,t_{k}$ as an input. In most circumstances, not all of the state variables of the system can be directly observed. For instance, in epidemiology, data on new confirmed cases of infected individuals is readily available, compared to data on other epidemiological statuses of the population. For this reason, we use the new cases reported over a certain period of time (e.g., days, weeks) from the model to formulate the inverse problem.

Let $y_{\mathrm{obs}}={\left\{y_{1,t},\cdots ,y_{n,t}\right\}}_{t=1}^{M}$, M∈N denote the observed new infected cases in each unit/period of time t∈1,2,⋯,M for the *n* patches. Let $Z_{i}\left(t,\theta \right)$ be the incidence in Patch *i* during (t−1,t], derived from the numerical solution of the forward problem (Equation 6) with parameter vector θ. The objective function of the inverse problem is thus defined as
(7)$$F\left(\theta \right)=\sum_{i=1}^{n}\sum_{t=1}^{M}\frac{{\left\{Z_{i}\left(t,\theta \right)-y_{i,t}\right\}}^{2}}{Z_{i}\left(t,\theta \right)}.$$

Consequently, the inverse problem becomes: compute $\hat{\theta }\in \Theta$ such that
(8)$$\hat{\theta }=\underset{\theta \in \Theta }{\mathrm{argmin}}F\left(\theta \right)\;s.t\;\Psi \left(\theta \right)=x,$$
where Θ is the feasible region of the parameter θ.

Problem (Equation 8) can be optimized using methods such as Landweber [116,117,118,119], faster methods such as Levenberg–Marquardt, or gradient-based numerical non-linear least squares minimization algorithms such as the Sequential Least SQuares programming algorithm (SLSQP) (see, e.g., [51,120]). Due to the complexity of the forward problem (Equation 6), the inverse problem (Equation 8) is complex, high-dimensional, and computationally intensive. As a consequence, when considering problem (Equation 8), it is difficult for gradient-based algorithms and/or other optimization techniques to determine the global minima, thereby easily failing to reach the optimal solution. As an alternative, we chose to solve problem (Equation 8) using a Genetic Algorithm (GA), a type of the diverse machine learning Evolutionary Algorithm (EA) that utilizes heuristic search and optimization methods. Besides their efficacy in the optimization of problems involving numerous parameters with large feasible regions (i.e., problems with increased dimensionality) and multiple local optima, GAs do not require gradient computations, which may be computationally challenging or altogether unavailable [121,122].

### 5.2. Results

As we considered four zones, we minimized Equation (Equation 7) as in (Equation 8) taking n=4. Besides estimating the epidemiological parameters ${⋃}_{i=1}^{4}\left\{\beta_{i},\kappa_{i},\gamma_{i}\right\}$, we also estimated the initial conditions ${⋃}_{i=1}^{4}\left\{E_{i,0},I_{i,0}\right\}$. Thus, from the deterministic inversion, we estimated a total of 20 parameters: $\theta ={⋃}_{i=1}^{4}\left\{\beta_{i},\kappa_{i},\gamma_{i},E_{i,0},I_{i,0}\right\}$.

Table 4 and Table 5 provide the estimated parameters, while Figure 6 and Figure 7 present the fitted incidence for the simulated and observed data, respectively. For the simulated incidence, the fit was visibly good, even though there was variation between the estimated and observed parameters (see Table 3) and a more important difference in the initial conditions. These disparities may be a result of the noise that was added to the simulated data, representing an important modification to the original simulated incidence (especially in the early stages of the outbreak). As in complex models, the situation where there exist different sets of parameters that would give an equally good fit to the zonal model incidences may have had an influence on the results.

As the estimated parameters from the GA had a good fit to the model incidence, we used them as a guide in specifying the initial sampling points for Stan and t-walk when performing Bayesian inference and uncertainty quantification (as detailed in the following section). The values were particularly important for t-walk to begin sampling as, without them, the algorithm does not converge and completely fails. In addition, as there are no studies providing the spaces (upper and lower bounds) for the initial conditions of the incubation and prevalence states parameters for Hermosillo, we used the estimates of the said parameters from the GA as a guide to set their boundaries in the Bayesian procedure.

## 6. Bayesian Inference

Deterministic inversion of epidemic models has gained prominence in terms of learning the trajectories of epidemics through the use of measured epidemiological data. However, deterministic inversion ignores inherent uncertainties due to imperfect data, stochasticity, model structural discrepancies, and even parameter uncertainties, which typically affect the epidemic model. In order to make robust, valid, and reliable decisions regarding the management, prediction, and forecasting of current and future epidemics, good treatment and quantification of such uncertainties is crucial. In this section, we use Bayesian techniques to quantify the uncertainties that characterize the epidemic model (Equation 1), its parameters, and the observed Hermosillo COVID-19 infection data, as well as providing credible intervals for the model output.

Over the past few decades, Bayesian inference has become an important approach, as it provides an optimal probability platform for learning unknown parameters of a system given observed data. A Bayesian framework learns the trajectories of the unknown parameters by updating their prior distributions to the posterior distribution. Such prior updates and exploitation of the parameter space of the posterior distribution can be achieved using standard Monte Carlo Markov Chain (MCMC) methods. Moreover, MCMC methods provide approximate estimates of intractable posterior distributions, which often characterize the modeling of real-world phenomena. The estimation of such intractable posterior distributions has been a long-standing challenge in the realm of Bayesian inference [123]. For this research, we use t-walk and Stan, which are MCMC-based algorithms, to simulate the posterior distribution. A detailed account of how the Bayesian framework is treated is presented in this section.

### 6.1. Building the Posterior Distribution

In Bayesian theory, the observed data $y_{\mathrm{obs}}$ are fixed, while the state variable x and the parameter θ are considered as random variables. In this setting, Bayes’ theorem defines the posterior distribution as
(9)$$\pi \left(\theta |y_{\mathrm{obs}}\right)\propto p\left(y_{\mathrm{obs}}|\theta \right)\pi \left(\theta \right),$$
where π(θ) (called the prior distribution) codifies our belief about the unknown parameter θ before the data were observed and $p\left(y_{\mathrm{obs}}|\theta \right)$ (called the likelihood distribution) codifies all the information available regarding how the observed data $y_{\mathrm{obs}}$ were obtained. From the Bayesian perspective, (Equation 9) constitutes an inverse problem which may be solved by using the *Maximum a posteriori* (MAP), an argument that maximizes the posterior ($\arg\;\max_{\theta \in \Theta }\pi \left(\theta |y_{\mathrm{obs}}\right)$), the posterior mean ($E\left[\pi \left(\theta |y_{\mathrm{obs}}\right)\right]$), or posterior median as the optimal value $\hat{\theta }$ of θ.

As we consider count data in this study—which, by their very nature, are events that occur within a given period of time—a Poisson process would be a natural and meaningful starting point for modeling and performing inference on the observed cases. However, count data, especially epidemiological observations, usually exhibit more variation than is implied by the Poisson distribution, pointing to inherent over-dispersion in the data. Such variations may be due to sampling, aggregation, environmental variability, or a combination of these factors, causing count data to have inherent over-dispersion. Therefore, it is not possible that count epidemiological data present an equal mean-variance relationship, as is assumed by the Poisson distribution. In fact, ref. [124] posited that most commonly used approaches assume a quadratic mean-variance relationship. The mean-variance relationship in modeling count data, such as epidemiological data, can be adequately described by the negative-binomial distribution, given that it includes an additional parameter that permits the variance to be larger than the mean [125,126,127]. Moreover, the negative binomial distribution is a mixture of Poisson and Gamma distributions [127,128], indicating that the Poisson distribution is still involved in the modeling process, even when the negative binomial distribution is used.

We assume that the observed data follow the negative binomial distribution with one dispersion parameter ν and write y∼NB(ν,p), where *p* is the probability of success. This distribution has the form
(10)$$p\left(y|\mu ,\theta \right)=\left(\begin{array}{c} y+\nu -1\\ \nu \end{array}\right){\left(\frac{\mu }{\mu +\nu }\right)}^{\nu }{\left(\frac{\nu }{\mu +\nu }\right)}^{y},$$
where
$$\mu =E\left(Y\right)=\frac{\nu p}{1-p}\;\mathrm{and}\;\mathrm{Var}\left(Y\right)+\frac{\mu^{2}}{\nu }=\mu +\rho \mu^{2}\;\mathrm{with}\;\rho =\frac{1}{\nu }.$$

It is worth noting that $\rho \mu^{2}>0$ is the additional variance allowed by the Negative Binomial distribution with respect to the Poisson distribution, and that the parameter ρ can be viewed as controlling the dispersion of the observed data.

We acknowledge that there are other forms of the Negative binomial distribution; for example, ref. [49] achieved success in modeling COVID-19 cases and hospital demand in Mexico using a Negative binomial distribution with two over-dispersion parameters. This distribution was proposed by [124]. However, in a model with numerous parameters to estimate—as is the case for the model considered here—it is our view that additional parameters would further increase the dimensionality of the already complex optimization problem.

We now detail the construction of the posterior distribution for the present multi-patch model study. Using the observed data and the variable θ described above, we assume that the observed cases in each patch follow a negative-binomial distribution with over-dispersion parameter (i.e., the number of failures before first success) $\nu_{i}$ and success probabilities
$$p_{i}=\frac{\mu_{i}}{\mu_{i}+\nu_{i}},$$
where $\mu_{i}$ is the theoretical mean of the negative binomial distribution, which can be taken as the incidence in patch *i*
(11)$$\mu_{i}\left(j,\theta \right)=Z_{i}\left(j,\theta \right),\;i=1,2,\ldots ,n\;j=1,2,\cdots ,M.$$

With this setting in mind, we have that, if $y_{i,j}$ represents the incidence in Patch *i* observed in day *j* with theoretical mean $\mu_{i}\left(j,\theta \right)$, as in (Equation 11), then we have
$$y_{i,j}|\theta ∼NB\left(\nu_{i},\frac{\mu_{i}\left(j,\theta \right)}{\mu_{i}\left(j,\theta \right)+\nu_{i}}\right).$$

Then, the likelihood from Patch *i*, i=1,2,⋯,n, corresponds to
$$\begin{array}{cc} p_{i}\left(y_{i}=\left(y_{i,1},\cdots ,y_{i,M}\right)|\theta \right) & =\prod_{j=1}^{M}\frac{\Gamma (y_{i,j}+\nu_{i})}{y_{ij}!\Gamma \left(\nu_{i}\right)}{\left(\frac{\mu_{i}\left(j,\theta \right)}{\mu_{i}\left(j,\theta \right)+\nu_{i}}\right)}^{\nu_{i}}{\left(\frac{\nu_{i}}{\mu_{i}\left(j,\theta \right)+\nu_{i}}\right)}^{y_{i,j}}\\ \; & \propto \left(\prod_{j=1}^{M}\frac{\Gamma (y_{i,j}+\nu_{i})}{\Gamma (\nu_{i})}\right){\left(\frac{\mu_{i}\left(j,\theta \right)}{\mu_{i}\left(j,\theta \right)+\nu_{i}}\right)}^{M\nu_{i}}{\left(\frac{\nu_{i}}{\mu_{i}\left(j,\theta \right)+\nu_{i}}\right)}^{\sum_{j=1}^{M}y_{i,j}}. \end{array}$$

Hence, the likelihood of the combined data from the *n* patches is given as $p\left(y_{\mathrm{obs}}|\theta \right)=\prod_{i=1}^{n}p_{i}\left(y_{i}|\theta \right)$.

To establish the prior distribution, we define the joint prior distribution as the product of the marginal; that is, if we denote $\theta =\cup_{i=1,\ldots ,n,l=1,\ldots ,m_{i}}\theta_{il}$, where $\theta_{il}$ is the *ℓ* parameter associated with Patch *i*, then
(12)$$\pi \left(\theta \right)=\prod_{l=1}^{m_{1}}g\left(\theta_{1l}\right)\prod_{l=1}^{m_{2}}g\left(\theta_{2l}\right)\cdots \prod_{l=1}^{m_{n}}g\left(\theta_{nl}\right)=\prod_{i=1}^{n}\left[\prod_{l=1}^{m_{i}}g\left(\theta_{il}\right)\right],$$
where $m_{i}$ is the number of parameters for Patch *i*.

Thus, the posterior distribution of the parameters from all *n* patches can be obtained as the product of (Equation 9) and (Equation 12), as (Equation 9).

### 6.2. Results

From the deterministic inversion section, we can note that we are interested in estimating the transmission, incubation, and recovery rates $\beta_{i}$, $\kappa_{i}$ and $\gamma_{i}$ for each zone *i* (i=1,2,3,4), as well as the exposed and infected incidence initial conditions $E_{i,0}$ and $I_{i,0}$. For the Bayesian inference (also called probabilistic inversion), four over-dispersion parameters $\nu_{i}$ are included, related to each zone’s observed incidence data. The dispersion parameters can be loosely defined as the number of trials before we encounter the first successful infection in each zone. These parameters come with the negative binomial distribution, which we use as the distribution of the observed zonal incidences. Thus, we have a total of 24 parameters for the Bayesian inference in this section, which we must estimate:(13)$$\theta =\overset{4}{\underset{i=1}{⋃}}\left\{\beta_{i},\kappa_{i},\gamma_{i},E_{i,0},I_{i,0},\nu_{i}\right\}.$$

An important problem that has been the topic of recent research is the selection of adequate prior distributions, their corresponding hyper-parameters, and the delimitations of the parametric spaces of the parameters to be estimated [129,130,131]. In this research, for the epidemiological parameters, we chose prior distributions with 95% of their body falling within the parametric spaces (upper and lower bounds) determined from the COVID-19 literature, as detailed in Table 2.

In this way, we considered the following prior distributions for each of the parameters:$$\begin{array}{cc} \beta_{i} & ∼\mathrm{LogNorm}\left(\mu_{i},\sigma_{i}\right)=g\left(\beta_{i}\right),\\ \kappa_{i} & ∼\mathrm{Gamma}\left(4.5264,19.1006\right)=g\left(\kappa_{i}\right),\\ \gamma_{i} & ∼\mathrm{Gamma}\left(1.9826,3.6943\right)=g\left(\gamma_{i}\right),\\ E_{i,0} & ∼\mathrm{Unif}\left(a_{i},b_{i}\right)=g\left(E_{i,0}\right),\\ I_{i,0} & ∼\mathrm{Unif}\left(0,10\right)=g\left(I_{i,0}\right),\\ \nu_{i} & ∼\mathrm{Gamma}\left(0.7561,0.2714\right)=g\left(\nu_{i}\right). \end{array}$$

Furthermore, the hyper-parameters for the log-Normal prior distributions of the contact parameters were $\mu_{1}=-0.0529$, $\mu_{2}=-0.0975$, $\mu_{3}=-0.8562$, $\mu_{4}=-0.2583$, and $\sigma_{1}=\sigma_{2}=\sigma_{3}=\sigma_{4}=0.2803$; while those for the initial conditions of the exposed state were $a_{1}=50$, $b_{1}=100$, $a_{2}=70$, $b_{2}=100$, $a_{3}=15$, $b_{3}=50$, and $a_{4}=0$, $b_{4}=20$. These prior probability distributions, which we selected using the above-mentioned criteria, are versatile and allow for the assumption that some parameter values could occur with lower, equal, or higher probability densities.

To avoid generalizations, it is imperative to perform a diagnostic analysis regarding the adequacy and correctness of the selected prior distributions before sampling. These prior distributions should be coherent with our expectations and correspond to the domain knowledge of each parameter obtained from the deterministic inversion in the previous section. We desire priors that—in accordance with our deterministic inversion domain experience—permit every conceivable configuration of the data while excluding blatantly ludicrous scenarios. Figure 8 shows a graph of the selected prior distributions for the infection, incubation, and recovery parameters, vis-à-vis the corresponding posterior distributions. We can observe that, for most parameters, we allowed for low informative priors and the posterior distributions were within their respective prior probability support. This conforms with the suggestion of [132], who stated that the model should not be overly confined by the priors, which should instead be too wide to encompass a wide variety of situations and data that are incredibly improbable. We thus conclude that our selected prior distributions were adequate and capable of regularizing our estimates to avoid non-identifiability. We acknowledge that [131] also proposed an interesting criterion for selecting prior distributions together with their corresponding hyper-parameters, based on parametric intervals. Applying the assumption of independence of the parameters, the joint prior distribution is $\tilde{\pi }\left(\theta \right)=\prod_{i=1}^{4}g\left(\beta_{i}\right)g\left(\kappa_{i}\right)g\left(\gamma_{i}\right)g\left(E_{i,0}\right)g\left(I_{i,0}\right)g\left(\nu_{i}\right)$.

The likelihood for the four zonal incidences is calculated as
$$p\left(y_{\mathrm{obs}}|\theta \right)\propto \prod_{i=1}^{4}\left[\left(\prod_{j=1}^{M_{i}}\frac{\Gamma (y_{i,j}+\nu_{i})}{\Gamma (\nu_{i})}\right){\left(\frac{\mu_{i}\left(j,\theta \right)}{\nu_{i}+\mu_{i}\left(j,\theta \right)}\right)}^{M_{i}\nu_{i}}{\left(\frac{\nu_{i}}{\nu_{i}+\mu_{i}\left(j,\theta \right)}\right)}^{\sum_{j=1}^{M_{i}}y_{i,j}}\right],$$
where $M_{i}$ is the total number of days that the incidence is observed for each zone i=1,2,3,4, $y_{i,j}$ is the $j^{\mathrm{th}}$ observed incidence for zone *i*, and $\mu_{i}\left(j,\theta \right)$ is the incidence provided by model (Equation 1). Thus, the posterior distribution is given as $p(\theta |y_{\mathrm{obs}})=p\left(y_{\mathrm{obs}}\right|\theta )\tilde{\pi }\left(\theta \right)$.

The incidence values output from the epidemic model (Equation 1) were very sensitive and exhibited a lot of uncertainty in response to slight changes, especially in terms of the contact, incubation, and recovery parameters. As a consequence, it was challenging to provide a wide range of values as support for the posterior distribution in Stan and t-walk. We addressed this challenge by beginning with deterministic inversion of the epidemic model using GA, as discussed in the previous section. The results of this inversion are given in Table 5. Consequently, we used the point parameter estimates obtained from the GA as the initial sampling points for t-walk and Stan. Moreover, as there is no literature reporting the parametric spaces for the initial conditions of the incubation and prevalence states, we delimited their support in the posterior distribution to within certain neighborhoods of their point parameter estimates obtained from the GA.

After performing the prior predictive checks, using the t-walk package [133], we obtained posterior samples for each of the 24 parameters (Equation 13). MCMC methods, by definition, produce samples with high correlation, harboring some level of redundant information. One way of measuring the level of information redundancy in the posterior sample is through use of the effective sample size (ESS). Theoretically, the ESS is the sample size we would obtain if we independently sampled the posterior distribution. The ESS and the proportion of ESS (pESS) for the simulated samples are given in the last two columns of Table A1. Going by how low (high) the ESS (pESS) values are, we can tell that the simulated chains were highly correlated and, thus, the utilized MCMC method required a very high number of iterations. In other words, in order to obtain samples with insignificant auto-correlations, thinning at large lags should be conducted, which can remarkably reduce the chain size. To obtain chains with reasonable sample sizes for analysis, after running the MCMC method for 1,000,000 iterations, we discarded 360,000 burn-in samples and thinned the chains at lag 20.

The summary statistics of the generated samples are provided in Table A1 (in the Appendix A). The presented summary statistics include the mean, variance, 95% credible interval (CI), median, ESS, and pESS. From this table, we can conclude that, in general, all the samples exhibited low variability and that their means were close to the estimates obtained by the GA in the previous section. Furthermore, we can conclude that the means (and medians) were all within the relevant 95% CIs. The former and the latter observations can be confirmed from Figure 9, which shows the 95% CIs of the samples, the means, and the estimates from the GA. Based on the scales of the values in the intervals, we can conclude that the credible intervals (CIs) were generally short—an indication of good estimates.

Figure 10 and Figure 11 present the posterior predictive checks. These figures graph the incidence simulations of model (Equation 1) from 26 February 2020 to 17 October 2020. This period includes all of the dates within which the incidence data were observed (26 February 2020 to 6 September 2020) and 41 prediction dates (7 September 2020 to 17 October 2020). Figure 10 shows 30,000 model incidences (gray curves) from the first 30,000 MCMC iterations, the observed incidence, and the maximum a posteriori (MAP) model incidence. Here, the MAP model incidence—which is apparently covered by the gray curves—is the model incidence output obtained from the parameter values that maximize the posterior distribution. We can observe, from this figure, that: (1) Our fitted model produced simulations that are consistent with the observed incidence data; and (2) the simulated trajectories did not present diverse or variable changes. Indeed, the latter observation is consistent with our previous observation that there was low variability in the simulated samples.

Figure 11 shows the 95% posterior credible intervals (CIs), mean, median, and MAP of the simulated model incidences. From this figure, we can see that most of the observed training incidences for the four zones fell outside the 95% CI of the simulated model incidence, with the zone 1 and zone 3 simulated model incidences providing the narrowest 95% CIs. In contrast, zones 2 and 4 had larger 95% prediction CIs, covering more of the observed incidences. The zone 1 model prediction interval provided the worst coverage of the observed prediction incidences. The proportions of the observed prediction incidences covered by the 95% CIs of the simulated model incidence from t-walk are presented in Table 6. As the CIs for the four zones were generally not too wide and covered a small proportion of the training observed incidence, and considering the prediction observed incidences for zones 3 and 4, we can conclude that the point estimates from t-walk seem to be adequate; however, the uncertainty associated with the estimated parameters tended to be underestimated.

We can observe, from both Figure 10 and Figure 11, that the model predicted that there will still be some infection levels beyond 6 September 2020—the date when we observed the last incidence. In fact, the figures reveal that, in the four zones, some individuals will still be infected up until 17 October 2020, with the median and MAP incidences being within the 95% CI.

After checking the adequacy of the prior distributions with respect to the posterior distribution and sampling, using the t-walk package, we also performed sampling of the posterior distribution using the Stan package. Stan is a Hamiltonian Monte Carlo (HMC)-based package designed for Bayesian statistical analysis; we refer the reader to [134] for more details about the software and [132,135] for insight into how the package can be used to perform inference and uncertainty quantification for compartmental epidemic models.

For Stan, we used the same support for the prior distribution as for t-walk, except for the incubation initial conditions. Using the same support for the incubation initial conditions in t-walk led to a very low overall proposal acceptance rate. As such, we used a different, shorter support range for all the incubation initial conditions prior distributions in t-walk, in order to obtain a reasonable proposal acceptance rate for the algorithm. However, in Stan, the support range for the same parameters were larger, as Stan is more robust. For the over-dispersion parameters, we used the transformed versions and sampled their inverses in Stan.

For the model at hand, we ran one chain with 1000 iterations for each of the 24 parameters and took the first 300 iterations as burn-in samples. The number of iterations was not the same as those obtained with t-walk, as Stan requires longer periods of time than t-walk to complete each iteration. However, from the analysis, we observed that the chains were importantly less correlated, and we thus obtained bigger effective sample sizes. Figure 12 and Figure 13, respectively, show the trace plots and the posterior densities for the samples of the estimated parameters. We obtained unit Rhat values from the Stan summary statistics output, an indication that all but two of the transitions converged. Additional Stan statistics—particularly, the diagnosis of the pairs plot of a subset of the estimated parameters (see Figure 14)—confirmed the convergence of all but two of the transitions of the chain during sampling.

Figure 15 shows the 95% CIs of the posterior distributions obtained from Stan. All the parameters, except for $\beta_{1}$, $\gamma_{1}$, $\gamma_{2}$, and $\gamma_{3}$, had short CIs. It is worth noting that (just as in t-walk) the samples of the inverse of the over-dispersion parameters $\nu_{i}$ from Stan had very narrow intervals.

The pair plots in Figure 14 show the posterior distributions of the infection and recovery parameters (diagonal figures) and scatter plots (off-diagonal figures) presenting the relationships between each of the samples. As there were many (24) parameters to estimate, it was not appealing to create a joint pairs plot for all of them. Inasmuch as this is the ideal procedure, doing this would lead to a figure with very tiny, non-visualizable figure entries. Thus, we chose to pair the infection and recovery rates (see Figure 14), as the scatter plots of these parameters exhibited interesting relationships. For instance, we can observe from the figure that there were strong linear positive posterior correlations between $\beta_{1}$ and $\gamma_{1}$, $\beta_{2}$ and $\gamma_{2}$, and $\beta_{4}$ and $\gamma_{4}$. In contrast, the infection parameters for the four zones did not exhibit any important relationships among themselves. This was also the case for the recovery parameters.

The summary statistics of the samples obtained from Stan are shown in Table A2. These statistics were slightly different from those obtained with t-walk, as displayed in Table A1. The differences in the statistics of the prevalence initial conditions and the over-dispersion parameters were obvious, as we have already mentioned that, besides using different support ranges for the distributions of the prevalence initial conditions, we transformed and sampled the inverses of the over-dispersion parameters $\nu_{i}$, i=1,2,3,4. Otherwise, the slight differences in the statistics of the other parameters could have been due to the difference in how the t-walk and Stan packages perform sampling. Notably, these differences would not lead to a different conclusion regarding the potency of COVID-19 infections in Hermosillo in 2020. The statistics in Table A2 reveal that the point estimates for the parameters were within the 95% CIs, as was the case for the estimated parameters in t-walk, as shown in Table A1.

The differences in the summary statistics of the samples obtained from both packages are further reflected in Figure 11 and Figure 16, which show the posterior predictive check figures for t-walk and Stan, respectively. We can observe that the mean, median, and observed incidences for each zone were all covered by the respective 95% CIs. In comparison, we can observe, from the t-walk output in Figure 11, that the 95% CI of the epidemic model incidences did not cover some observed daily incidences, while the same interval with Stan covered all of the training observed zonal incidences. We can thus conclude that the estimated parameters from Stan capture a more realistic uncertainty level, compared to those estimated by t-walk. Figure 11 and Figure 16 reveal that the COVID-19 infections in the four zones—and consequently, the whole of Hermosillo—peaked in mid-July of 2020, with a prediction of continued infections beyond 6 September 2020 (i.e., the last date of observed training daily incidences).

Table 2 provides the ranges for the parameters of the model, as obtained with reference to the COVID-19 literature. Our results from the GA, t-walk, and Stan packages (see Table 5, Table A1 and Table A2, respectively) indicated that the point estimates of the infection and incubation parameters ($\beta_{i}$ and $\kappa_{i}$, i=1,2,3,4) that best fit the incidences for the four zones were within the ranges derived from the literature. However, this was not the case for the recovery parameters ($\gamma_{i}$), whose estimates were generally close to the one-day recovery period average for the four zones. This result is not surprising, as our incidence data for the four zones were inflated by a large under-reporting factor. This inflation, besides indicating a possible lack of systematic testing for COVID-19 in Hermosillo, also accounted for the existence of sub-clinical and asymptomatic COVID-19 patients in the four zones under study. Going by how well the models fit the data (see Figure 7, Figure 10, Figure 11 and Figure 16), we believe that accounting for incidence under-reporting by a factor of 15 (as reported by [86]) is realistic.

We further evaluated the prediction capability of the multi-patch model by checking the proportion of observed zonal prediction incidences that were within the 95% CIs of the zone model incidences. Table 6 shows these proportions obtained from Stan (see Figure 16) and t-walk (see Figure 11). From this table, we can see that Stan produced 95% prediction bands covering more than 50% of the observed prediction incidences for all of the zones. On the other hand, t-walk provided 95% CIs large enough to cover less than 50% of observed prediction incidences for all the four zones, with zone 1 providing the narrowest simulated model incidence CIs, covering only 22.22% of the observed prediction incidence. In effect, we can conclude that Stan outperforms t-walk, with regard to the percentage of covered observed daily prediction incidence for all of the considered zones.

### 6.3. Comparison to a Single-Patch Model

To assess the improvement obtained by using a more complex model, compared to a typical single population model, we fit a single-patch model (the global SEIRS model) and compared its predictive performance against the multi-patch model with mobility. The systems of differential equations forming the single-patch model are presented in Equation (Equation 14).
(14)$$\left\{\begin{array}{c} \dot{S}=\Lambda -\beta SI/N-\mu S+\tau R\\ \dot{E}=\beta SI/N-\left(\kappa +\mu \right)E\\ \dot{I}=\kappa E-\left(\gamma +\phi +\mu \right)I\\ \dot{R}=\gamma I-\left(\mu +\tau \right)R. \end{array}\right.$$

We run the single-patch model using similar birth, natural death, and disease-induced mortality rate values as in the multi-patch model. We estimated the infection (β), incubation (κ), and recovery (γ) rates, as well as the initial conditions $E_{0}$ and $I_{0}$, for the single-patch model. The prior distributions of these parameters and the likelihood of the global observed incidence remained the same as in the multi-patch model. The initial population was taken as the sum of the population sizes for zones 1–4, as shown in Figure 1; that is, $N_{0}=\sum_{i=1}^{4}N_{i}\left(0\right)=858,812$.

We used the observed global incidence in the period of 26 February 2020 to 6 September 2020 in order to train the global single-patch model, then used the incidence in the period of 7 September 2020 to 17 October 2020 to assess the prediction efficiency of the model. Table 7 provides the summary statistics of the estimated parameters. For comparison with the multi-patch model, we summed the predicted mean and median incidences for all of the zones and computed the efficiency measures. The efficiency measures used in this case were the Root Mean Squared Error (RMSE) and Mean Absolute Percentage Error (MAPE), defined as
$$\mathrm{RMSE}=\sqrt{\frac{1}{n}\sum_{t=1}^{n}{\left(Y_{t}-{\hat{Y}}_{t}\right)}^{2}}\;\mathrm{and}\;\mathrm{MAPE}=\frac{100}{n}\sum_{t=1}^{n}\left|\frac{Y_{t}-{\hat{Y}}_{t}}{Y_{t}}\right|,$$
where *n* is the total number of the predicted incidences, $Y_{t}$ is the actual observed incidence, and ${\hat{Y}}_{t}$ is the predicted incidence (which, in this case, is the mean and median incidences for both the multi-patch and global single-patch models).

Figure 17 shows the global single-patch predicted incidences together with the 95% credible incidence intervals, and Table 8 provides the efficiency measures of the predictions for both the multi- and single-patch models. From this table (and, indeed, by comparing Figure 16 and Figure 17), we can see that the multi-patch model outperformed the global single-patch model, as it obtained predicted incidences with lower RMSE and MAPE values. Moreover, Figure 16 reveals that a bigger percentage of the observed prediction and model-predicted incidences from the multi-patch model were within the 95% credible incidence interval. Thus, we can conclude that, in terms of modeling COVID-19 in Hermosillo, using the multi-patch model with mobility represents an important improvement, when compared to the single-patch model.

## 7. Conclusions

In this work, we utilized three techniques to estimate the parameters and perform inference and uncertainty quantification on a multi-patch epidemic model including mobility, residency, and demography factors. An analysis was conducted using mobile phone GPS data and confirmed COVID-19 cases in four zones of Hermosillo, Mexico. The model comprises two sets of parameters—20 mobility and residence time parameters and 20 epidemiological parameters (including the incubation and prevalence initial conditions)—which were all estimated.

The complexity of epidemic models makes them susceptible to the problem of parameter non-identifiability, a complication which we addressed by using two distinctive data sets and different estimation techniques to estimate the two sets of parameters separately. In the first step, we used a Brownian Bridge model and mobile phone GPS data to estimate the first set of 20 mobility and residence times for the four zones in Hermosillo. The next set of the model epidemiological parameters were estimated using confirmed positive COVID-19 cases and two estimation techniques: deterministic inversion using GA and probabilistic inversion, and the latter using two different MCMC methods implemented through the t-walk and Stan statistical packages.

The incidence outputs of the epidemic model were very sensitive to slight changes in its parameters, especially the infection, incubation, and recovery rates. This necessitated the use of the deterministic inversion technique as the first step of estimation, which provided us with an idea of the possible parametric values and model incidences that best fit the observed incidences. This first step of estimation thus allowed for the determination of the scope of the parametric space within which the support of the posterior distribution and the initial sampling values used in the probabilistic inversion stage (with t-walk and Stan) could be delimited.

We are not concerned with a detailed comparison of t-walk and Stan, as there could be many factors affecting their performance, which undoubtedly vary among different models. Some of these factors may include the chosen parametrization of the model, the prior distribution, and initial values from where the sampling starts. However, we should mention that some studies have established that it may be difficult to use MCMC-based inference tools to model systems of ODEs, as the relevant likelihood functions may contain several local minima that causes them to violate the standard regulatory conditions. In these cases, Stan offers high statistical efficiency and, in complex scenarios, may produce more accurate results when compared to t-walk. Indeed, Stan performed better than t-walk for the presented epidemic model, as the posterior distributions were characterized by highly correlated parameter spaces. We observed this fact in the present study based on the large ESS proportions obtained from Stan, compared to those obtained from t-walk.

The estimation results obtained in this study indicated that the infection and incubation rates for the four considered zones in Hermosillo were within the ranges postulated in various studies in the COVID-19 literature. However, the recovery parameters fell outside the estimated ranges reported in other COVID-19-based studies. This deviation was, however, justified as possibly resulting from the inflation of incidences in the four zones by a factor of 15, in order to account for under-reporting. Regarding the estimated parameters, the model presented a good fit to COVID-19 infections in the four zones and, consequently, the whole city of Hermosillo; furthermore, under the model, the cases also peaked in mid-July of 2020.

The epidemic model and the results obtained in this study can be useful to public health officials as they not only provide better predictions, but as mobility parameters that can be modified to assess the effectiveness of different levels of mobility restrictions were explicitly introduced. These restrictions can be global or applied to only some patches. We believe that the detailed model and methods can be developed into a user-friendly tool to guide public health policies for the management and mitigation of emerging and re-emerging infectious diseases.

The epidemic model detailed in this article was confined within the limits of modeling a single strain of the COVID-19 virus. However, like many infectious diseases, multiple strains of COVID-19 have emerged as a result of its genetic mutation. For example, four mutants of the pathogen were detected within the first one and a half years of the pandemic. Therefore, it would be of great relevance to extend the proposed epidemic model to include the dynamics of multiple COVID-19 pathogenic strains. Such an extension could lead to better understanding and management of future infectious outbreaks.

In addition, besides urban mobility, the epidemic model used in this research did not account for government intervention measures, such as vaccination and regional or national mobility. As the vaccination campaign in Mexico started in December 2020, this factor is absent; however, it would be desirable to introduce these (and other) mitigation strategies in future models, in order to model subsequent outbreaks of COVID-19 and other infectious diseases.

It is worth noting that the assumptions for the proposed model include those for the ODEs to describe the dynamics within each patch, while ODEs can be effective in approximating dynamics in many scenarios, there are cases where—even with homogeneous mixing—the mathematical system may fail to accurately represent the behavior of a small population.

One reason for this is the assumption of continuous variables and smooth transitions inherent to ODEs. In small populations, discrete effects and stochastic events can have a significant impact on the dynamics. These discrete effects, such as random fluctuations or individual interactions, introduce variability that is not captured by the ODEs, leading to the model inaccurately reflecting the true behavior of the system.

To address this, one option for small spatial scales and patch sizes is to leverage computational techniques such as agent-based modeling. This approach captures the important variability that arises from individual interactions [136,137,138,139,140,141]. Additionally, in order to reduce the variability of simulations, relevant contact patterns produced by street or building layouts can be incorporated. At a more detailed level, modeling individual interactions as a network of contacts still allows for fitting or performing statistical inference using computer-intensive methods [55,142,143,144]. For these reasons, scaling up such methods to slightly larger patches can pose serious challenges that require careful considerations.

## Figures and Tables

**Figure 1 entropy-25-00968-f001:**
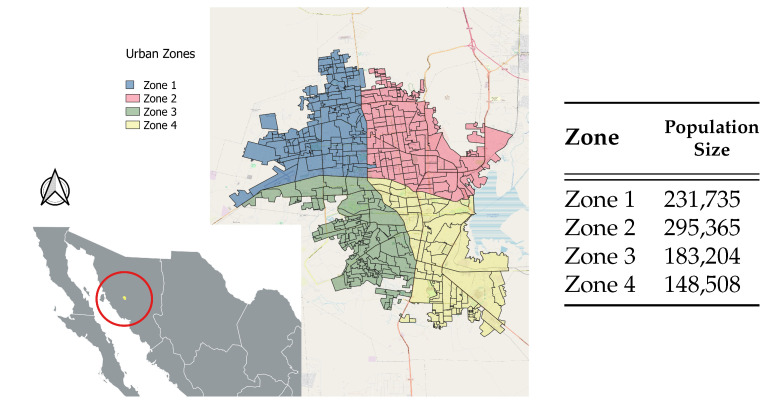
Data for Hermosillo city and Urban Zones. The city location within Mexico is identified as the yellow center of the red circle.

**Figure 2 entropy-25-00968-f002:**
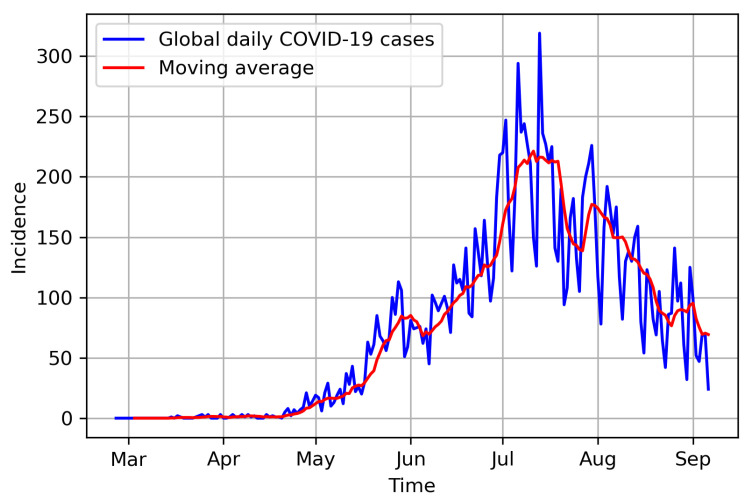
2020 COVID-19 daily cases in Hermosillo. Source: https://datos.COVID-19.conacyt.mx (accessed on 1 November 2022).

**Figure 3 entropy-25-00968-f003:**
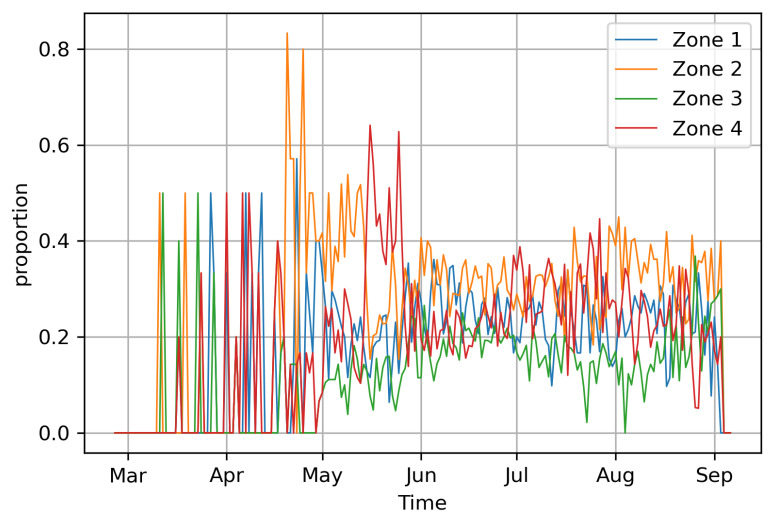
Proportions of zonal 2020 COVID-19 daily cases.

**Figure 4 entropy-25-00968-f004:**
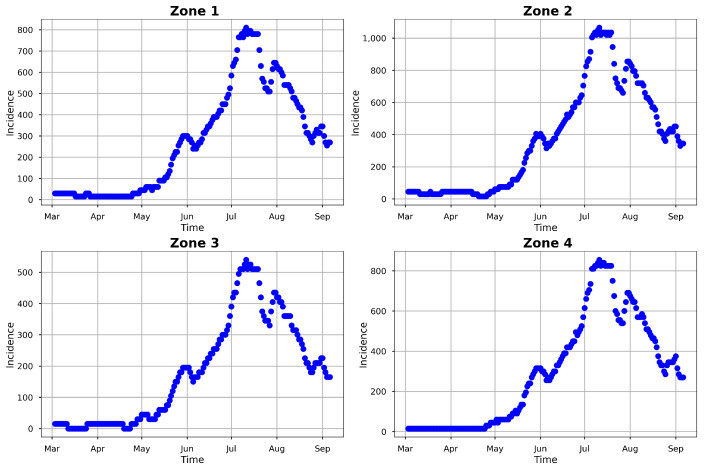
Observed 2020 COVID-19 daily cases in Hermosillo. The incidences (blue dots) are inflated by a factor of 15.

**Figure 5 entropy-25-00968-f005:**
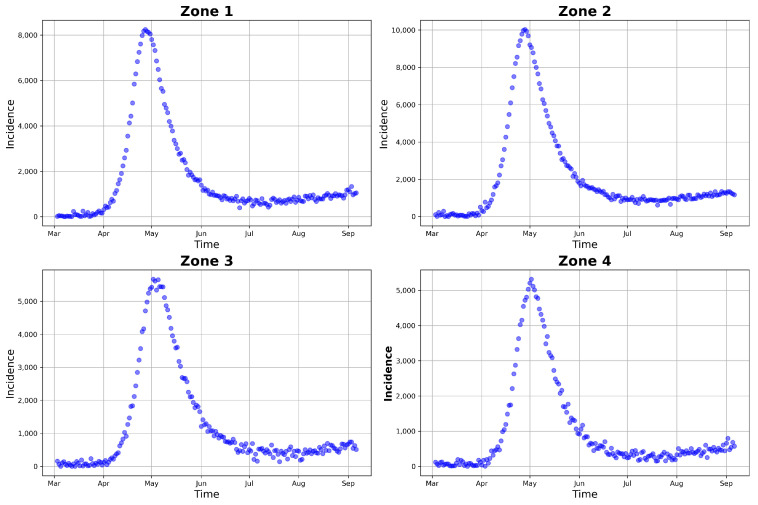
Simulated COVID-19 daily cases in Hermosillo. The daily model incidence (in blue) was simulated using the parameter values given in Table 3, together with the μ and ψ values provided in Table 2 and τ=1/180.

**Figure 6 entropy-25-00968-f006:**
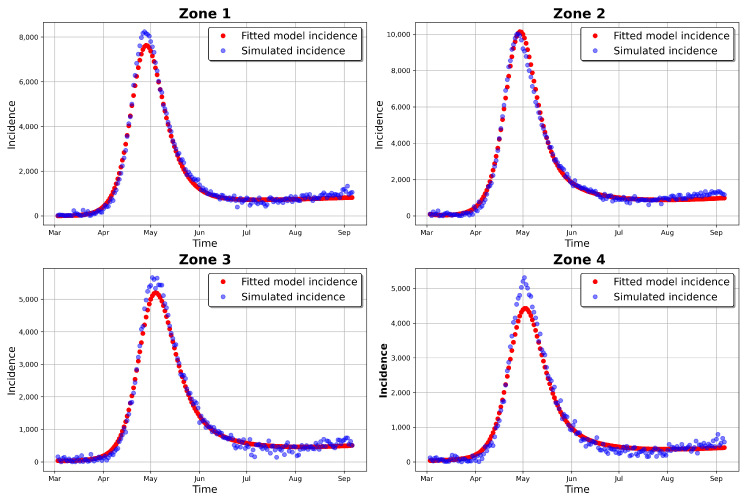
Fitted model incidence and simulated observed daily incidence values.

**Figure 7 entropy-25-00968-f007:**
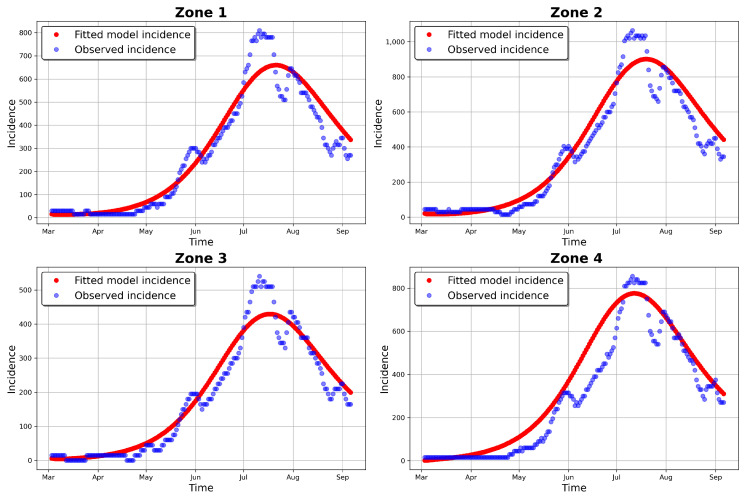
Fitted model incidence and observed 2020 daily incidence values.

**Figure 8 entropy-25-00968-f008:**
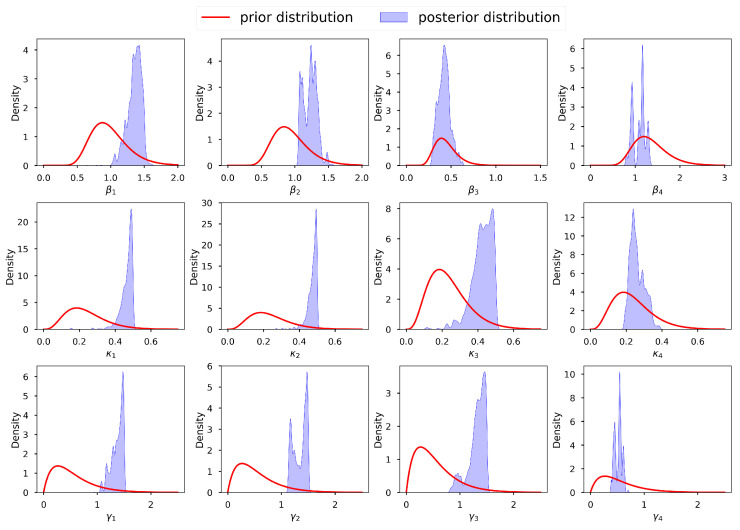
Prior distributions of the infection, incubation, and recovery rate parameters and corresponding posterior distributions obtained from t-walk.

**Figure 9 entropy-25-00968-f009:**
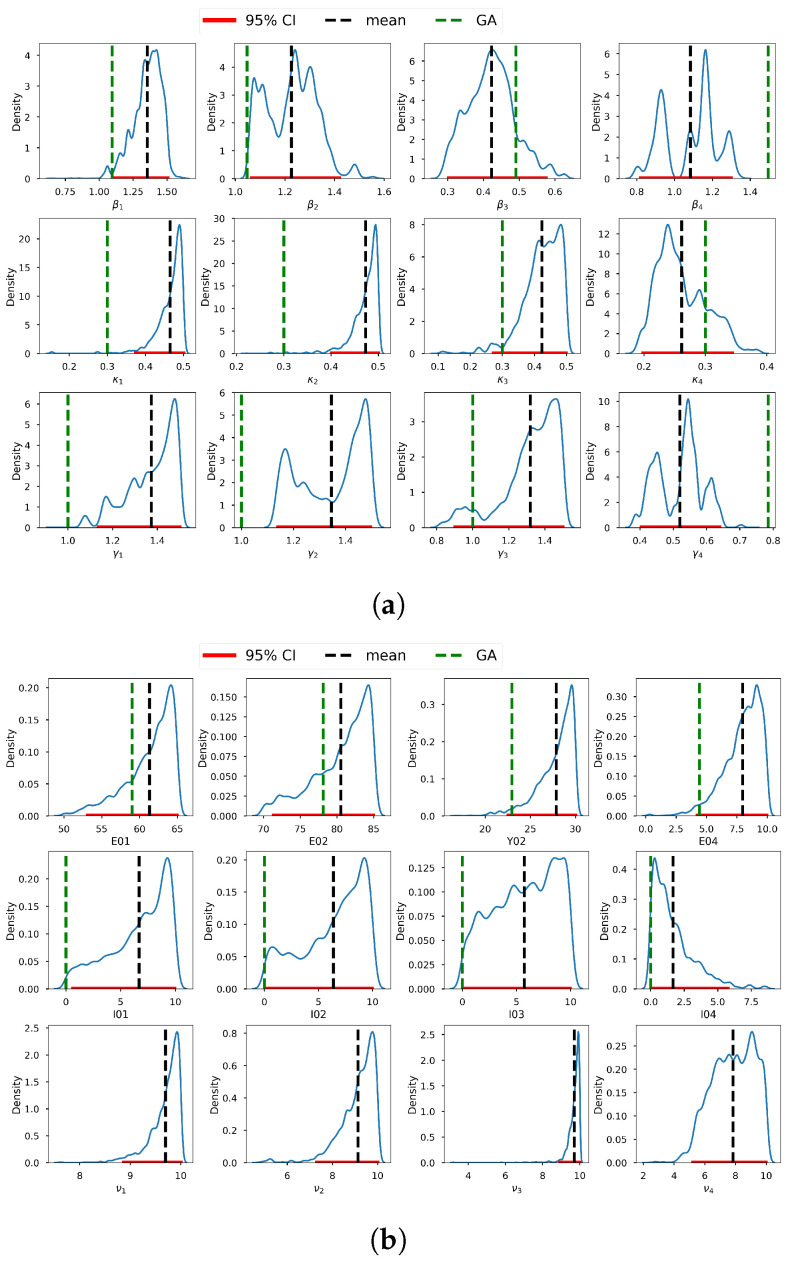
Credible intervals, means (from t-walk), and GA estimates of the incubation and recovery rates, exposed and prevalence initial conditions, and dispersion parameters: (**a**) Credible intervals, means, and GA estimates of the infection, incubation and recovery rates; and (**b**) credible intervals, means, and GA estimates of the exposed and prevalence initial conditions and the dispersion parameters.

**Figure 10 entropy-25-00968-f010:**
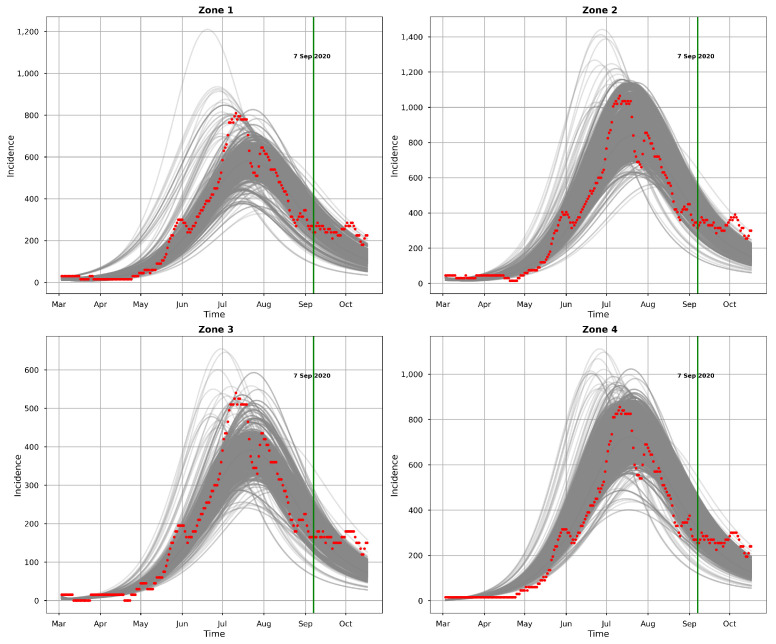
Observed (red points) and model (gray) incidences. The model incidences were obtained by solving the model using parameters from the first 30,000 MCMC iterations after burn-in. The observed incidences are for the year 2020.

**Figure 11 entropy-25-00968-f011:**
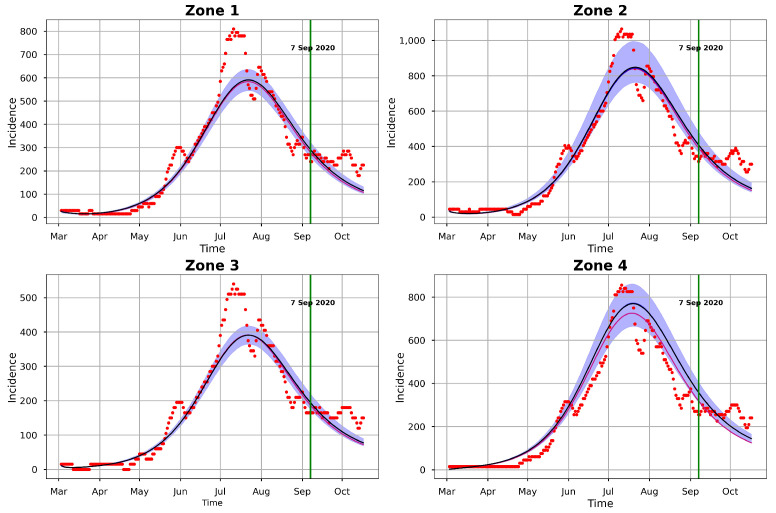
The mean (black), median (medium blue), MAP (medium violet red), and 95% CI (blue bands) of model incidence obtained from t-walk. The red points indicate observed daily incidences for the year 2020.

**Figure 12 entropy-25-00968-f012:**
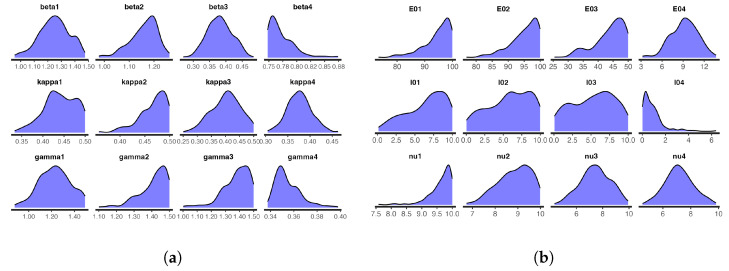
Posterior distributions of the infection, incubation, and recovery rates; exposed and prevalence initial conditions; and dispersion parameters obtained from Stan: (**a**) Infection, incubation, and recovery rate parameters; and (**b**) exposed and prevalence initial conditions and dispersion parameters.

**Figure 13 entropy-25-00968-f013:**
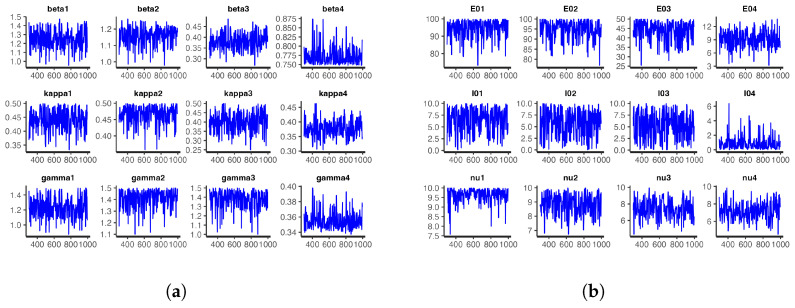
Trace plots of the samples of infection, incubation, and recovery rates; exposed and prevalence initial conditions; and over-dispersion parameters obtained from Stan: (**a**) Infection, incubation, and recovery rate parameters; and (**b**) exposed and prevalence initial conditions and the dispersion parameters.

**Figure 14 entropy-25-00968-f014:**
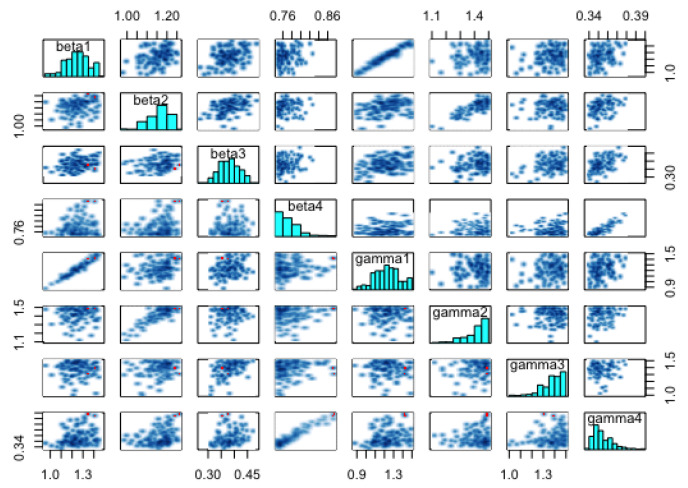
Pair plots for the infection samples and recovery rates obtained from Stan.

**Figure 15 entropy-25-00968-f015:**
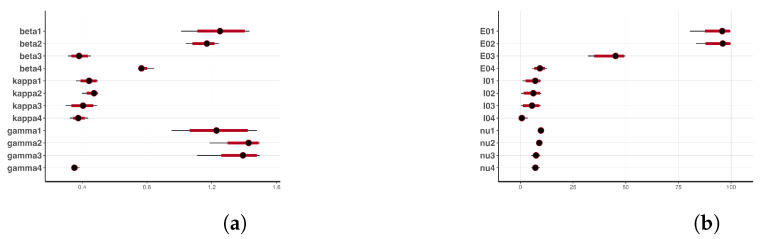
95% Credible intervals of the estimated parameters obtained from Stan: (**a**) infection, incubation, and recovery rates parameters; and (**b**) exposed and prevalence initial conditions and the dispersion parameters.

**Figure 16 entropy-25-00968-f016:**
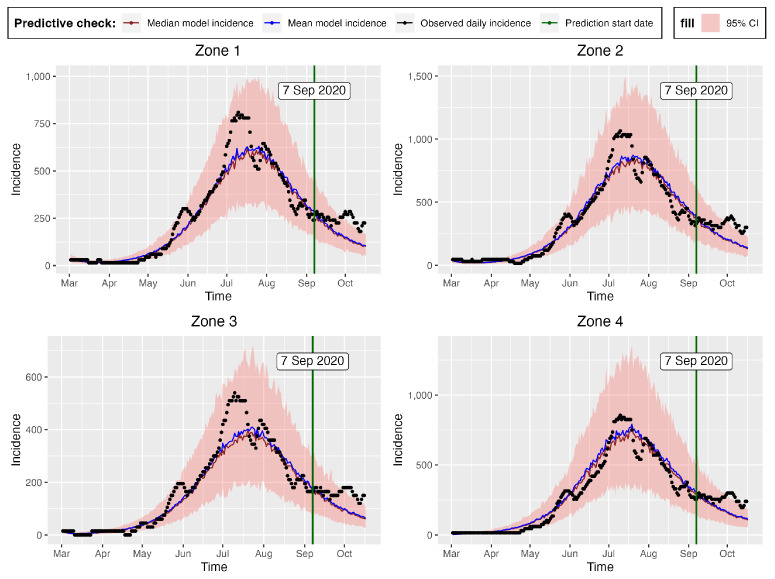
Observed daily incidence for the year 2020, posterior incidence predictions, and 95% uncertainty intervals obtained from Stan.

**Figure 17 entropy-25-00968-f017:**
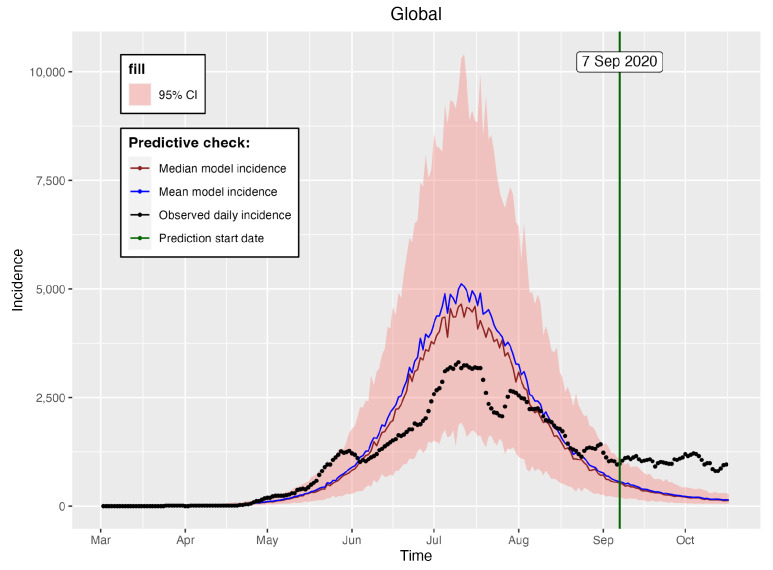
Global single-patch observed and predicted mean and median incidences together with the 95% CI. The observed daily incidences are for the year 2020.

**Table 1 entropy-25-00968-t001:** Parameter descriptions.

Parameter	Description
$\alpha_{i}$	The proportion of individuals that leave their residence patch *i*
$p_{ij}$	The proportion of time that an individual from patch *i* spends
	in patch *j*, given that the individual leaves their residence patch
$\Lambda_{i}$	Recruitment of Susceptible individuals in Patch *i*
$\beta_{j}$	Instantaneous risk of infection in Patch *j*
$\mu_{i}$	Per capita natural death rate in Patch *i*
$\gamma_{i}$	Per capita recovery rate of individuals in Patch *i*
$\tau_{i}$	Per capita loss of immunity rate in Patch *i*
$\psi_{i}$	Per capita disease-induced death rate in Patch *i*
$\kappa_{i}$	Per capita rate at which the exposed individuals in patch *i* become infectious

**Table 2 entropy-25-00968-t002:** Epidemiological parameter ranges/values and their associated references.

Parameters	Range/Value	Literature
β	0.0616–1.68	[88,89,90]
1/κ	2–14	[88,91,92,93,94]
1/γ	5–15.6	[49,88,90,95,96]
μ	0.06/(1000×365)	[97]
ψ	(0.0047 + 0.0030)/2	[98]
1/τ	90–300	[99,100,101,102,103,104,105,106,107,108]

**Table 3 entropy-25-00968-t003:** Parameter values for Model (Equation 1) used for simulation considering the observed daily incidence values (see Figure 5).

Epidemiological		Initial Conditions	
**Parameter**	**Value**	**Parameter**	**Value**
$\beta_{1}$	1.3000	$E_{1,0}$	10
$\beta_{2}$	1.4000	$I_{1,0}$	1
$\beta_{3}$	0.9500	$Y_{1,0}$	1
$\beta_{4}$	0.8000	$E_{2,0}$	20
$\kappa_{1}$	0.0833	$I_{2,0}$	2
$\kappa_{2}$	0.0714	$Y_{2,0}$	2
$\kappa_{3}$	0.0741	$E_{3,0}$	5
$\kappa_{4}$	0.1000	$I_{3,0}$	0
$\gamma_{1}$	0.1667	$Y_{3,0}$	0
$\gamma_{2}$	0.1429	$E_{4,0}$	5
$\gamma_{3}$	0.1818	$I_{4,0}$	0
$\gamma_{4}$	0.2000	$Y_{4,0}$	0

**Table 4 entropy-25-00968-t004:** Model (Equation 1) parameter values estimated from deterministic inversion using a Genetic Algorithm (GA). The values were used to obtain the fitted simulated model incidence shown in Figure 6.

Epidemiological		Initial Conditions	
**Parameter**	**GA**	**Parameter**	**GA**
$\beta_{1}$	0.8322	$E_{1,0}$	0.0000
$\beta_{2}$	0.7473	$I_{1,0}$	0.0000
$\beta_{3}$	0.5259	$Y_{1,0}=E_{1,0}+I_{1,0}+R_{1,0}$	0.0000
$\beta_{4}$	1.4354	$E_{2,0}$	200.0000
$\kappa_{1}$	0.0829	$I_{2,0}$	126.4590
$\kappa_{2}$	0.5334	$Y_{2,0}=E_{2,0}+I_{2,0}+R_{2,0}$	326.4590
$\kappa_{3}$	0.1983	$E_{3,0}$	200.0000
$\kappa_{4}$	0.2115	$I_{3,0}$	0.0000
$\gamma_{1}$	0.0357	$Y_{3,0}=E_{3,0}+I_{3,0}+R_{3,0}$	200.0000
$\gamma_{2}$	1.0000	$E_{4,0}$	200.0000
$\gamma_{3}$	0.4248	$I_{4,0}$	8.4447
$\gamma_{4}$	1.0000	$Y_{4,0}=E_{4,0}+I_{4,0}+R_{4,0}$	208.4447

**Table 5 entropy-25-00968-t005:** Parameter values for Model (Equation 1) estimated from deterministic inversion using a Genetic Algorithm (GA). The values were used to obtain the fitted model and observed daily incidence shown in Figure 7.

Epidemiological		Initial Conditions	
**Parameter**	**GA**	**Parameter**	**GA**
$\beta_{1}$	1.0950	$E_{1,0}$	58.9925
$\beta_{2}$	1.0473	$I_{1,0}$	0.0000
$\beta_{3}$	0.4904	$Y_{1,0}=E_{1,0}+I_{1,0}+R_{1,0}$	58.9925
$\beta_{4}$	1.4951	$E_{2,0}$	78.1313
$\kappa_{1}$	0.3000	$I_{2,0}$	0.0000
$\kappa_{2}$	0.3000	$Y_{2,0}=E_{2,0}+I_{2,0}+R_{2,0}$	78.1313
$\kappa_{3}$	0.3000	$E_{3,0}$	22.9404
$\kappa_{4}$	0.3000	$I_{3,0}$	0.0000
$\gamma_{1}$	1.0000	$Y_{3,0}=E_{3,0}+I_{3,0}+R_{3,0}$	22.9404
$\gamma_{2}$	1.0000	$E_{4,0}$	4.4192
$\gamma_{3}$	1.0000	$I_{4,0}$	0.0000
$\gamma_{4}$	0.7853	$Y_{4,0}=E_{4,0}+I_{4,0}+R_{4,0}$	4.4192

**Table 6 entropy-25-00968-t006:** Proportion of the observed prediction incidence for considered zones covered by the model incidence CIs obtained from t-walk and Stan.

	% of Observed Prediction Incidence Covered by CIs	
Zone	Stan	t-walk
Zone 1	55.56	22.22
Zone 2	55.56	44.44
Zone 3	58.33	30.56
Zone 4	63.89	38.89

**Table 7 entropy-25-00968-t007:** Statistics of the single-patch model parameters estimated using probabilistic (Stan) inversion.

	Probabilistic Inversion (Stan) Statistics								
Parameter	Mean	SD	$q_{0.025}$	$q_{0.250}$	$q_{0.5}$	$q_{0.750}$	$q_{0.975}$	ESS	pESS
β	1.0139	0.0972	0.8168	0.9503	1.0191	1.0843	1.1701	244	0.0122
κ	0.8690	0.0847	0.6911	0.8140	0.8798	0.9436	0.9917	301	0.0151
γ	0.8569	0.0862	0.6797	0.8015	0.8640	0.9191	0.9936	244	0.0122
$E_{0}$	1.2060	0.8259	0.0497	0.5090	1.0952	1.7977	2.8544	296	0.0148
$I_{0}$	1.6101	0.7787	0.1173	1.0340	1.6587	2.2338	2.9343	237	0.0119
ν	4.0792	0.5313	3.0715	3.7108	4.0322	4.4569	5.1152	264	0.0132

**Table 8 entropy-25-00968-t008:** Efficiency measures for the predicted mean and median incidences for the multi- and single-patch epidemic models.

	Multi-Patch		Single-Patch	
Efficiency Measure	Mean	Median	Mean	Median
RMSE	395.61	418.61	773.98	796.06
MAPE	34.40	37.06	73.86	76.10

## Data Availability

The global Hermosillo COVID-19 data are publicly available from the CONACyT website https://datos.COVID-19.conacyt.mx (accesed on 1 November 2022). The GPS mobility and zonal COVID-19 data utilized in this study are available upon request from the corresponding author. These data sets are not publicly accessible, due to the inclusion of private and personal individual information. The geo-referenced COVID-19 data provided by the Secretaría de Salud del Estado de Sonora (SSA) are protected by confidentiality agreements and in accordance with the laws of personal data of Mexico, as they contain sensitive personal information such as medical records and residence locations. Detailed information regarding the processing, analysis, and visualization of this data can be found at https://www.mat.uson.mx/web/index.php/coronavirus-COVID-19/ (accesed on 1 November 2022), specifically on the official website of the Mathematics Department at UNISON. The codes used in this research are publicly available at https://github.com/AlbertAkuno/Inference (accesed on 1 November 2022).

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
