# Peer review of "Inference on a Multi-Patch Epidemic Model with Partial Mobility, Residency, and Demography: Case of the 2020 COVID-19 Outbreak in Hermosillo, Mexico"

_entropy, 2023, doi:10.3390/e25070968_

Round 1

Reviewer 1 Report

Authors specified that the University signed an agreement with the government for the use of data on COVID-19 cases, but it is not clear whether and in what manner authors acquired the consent even for the use of GPS data. This is a relevant issue that should be specified befor the review process would be completed.

Minor editing of English language required.

Reviewer 2 Report

Overall, this paper reports interesting research on a covid-19 epidemic model that is well-done and the findings are interesting.

Recommendation for revision: Carefully review the text of the paper to make sure that, on first usage, some key terms are well described and defined: for example, single-patch, multi-patch, SEIRS.

Overall, the English is well done. It does, however, need a careful polishing edit. 

Reviewer 3 Report

It is a good work that is suggested to be accepted.  Minor editing of writing is suggested.

It is suggested to have minor checking on editing.

Reviewer 4 Report

The author proposed an intresting idea - to explore the mobility of the pandemic spread but on the other way around. Much work has been done on how mobility influences the pandemic but no one paid much attention to how badly we understand this mobility. While the model and data approach is not novel at all (A paper that even uses mobile data was published a year and a half ago), the results are novel and I believe this is more than enough to be considered for publication. Since I do not have any critics of the research itself, I suggest a minor revision in which the author should fix several shortcomings in their writing (which overall is done very well):

1. The authors should provide more discussion about the limitation of their model. In particular, the authors focused on single-strain mutation which is in practice quite rare and short-lived in real pandemics. I suggest the authors address this point and refer to possible future work that remady it. To help, here are some recent works in this direction:
A. doi: 10.1109/ACCESS.2022.3149956.
B. M. Fudolig and R. Howard, "The local stability of a modified multi-strain SIR model for emerging viral strains", PLoS ONE, vol. 15, no. 12, Dec. 2020.
C. A. Yusuf, S. Qureshi, M. Inc, A. I. Aliyu, D. Baleanu and A. A. Shaikh, "Two-strain epidemic model involving fractional derivative with Mittag–Leffler kernel", Chaos Interdiscipl. J. Nonlinear Sci., vol. 28, no. 12, Dec. 2018.
D. https://doi.org/10.1371/journal.pone.0260683
E. O. Khyar and K. Allali, "Global dynamics of a multi-strain SEIR epidemic model with general incidence rates: Application to COVID-19 pandemic", Nonlinear Dyn., vol. 102, no. 1, pp. 489-509, Sep. 2020.

2. If possible, I really suggest that the author share the code used for this research.

3. It is of interest if the author can discuss how robust their method is. Does it works similar for smaller spatial scales such as streets? buildings? rooms? If not, what "breaks" the proposed method and how future work can address it. Please make sure to include recent works of pandemics in which spatial scales.

The level of English is fine
